# Improve Aggressive Driver Recognition Using Collision Surrogate Measurement and Imbalanced Class Boosting

**DOI:** 10.3390/ijerph17072375

**Published:** 2020-03-31

**Authors:** Ke Wang, Qingwen Xue, Yingying Xing, Chongyi Li

**Affiliations:** 1College of Transportation Engineering, Tongji University, Shanghai 201804, China; kew@tongji.edu.cn (K.W.);; 2Key Laboratory of Road and Traffic Engineering of the State Ministry of Education, Tongji University, Shanghai 201804, China; 3Shanghai Key Laboratory of Rail Infrastructure Durability and System Safety, Tongji University, Shanghai 201804, China

**Keywords:** imbalanced class boosting, vehicle trajectory, collision surrogate measurement, driving aggressiveness

## Abstract

Real-time recognition of risky driving behavior and aggressive drivers is a promising research domain, thanks to powerful machine learning algorithms and the big data provided by in-vehicle and roadside sensors. However, since the occurrence of aggressive drivers in real traffic is infrequent, most machine learning algorithms treat each sample equally and prone to better predict normal drivers rather than aggressive drivers, which is our real interest. This paper aims to test the advantage of imbalanced class boosting algorithms in aggressive driver recognition using vehicle trajectory data. First, a surrogate measurement of collision risk, called Average Crash Risk (ACR), is proposed to calculate a vehicle’s crash risk. Second, the driver’s driving aggressiveness is determined by his/her ACR with three anomaly detection methods. Third, we train classification models to identify aggressive drivers using partial trajectory data. Three imbalanced class boosting algorithms, SMOTEBoost, RUSBoost, and CUSBoost, are compared with cost-sensitive AdaBoost and cost-sensitive XGBoost. Additionally, we try two resampling techniques with AdaBoost and XGBoost. Among all algorithms tested, CUSBoost achieves the highest or the second-highest Area Under Precision-Recall Curve (AUPRC) in most datasets. We find the discrete Fourier coefficients of gap as the key feature to identify aggressive drivers.

## 1. Introduction

Active safety systems, such as Forward Collision Warning (FCW), Dynamic Brake Support (DBS), and Autonomous Emergency Braking (AEB), have been widely offered by many automobile companies in today’s new car. Active safety systems use radar, camera, or ultrasonic sensors to detect vehicles, pedestrians, and obstacles. Once an imminent collision is detected, these systems will alert drivers or intervene to avoid crashes. In the meantime, with the increasing ability in driving behavior data collection, there is an emerging possibility to take advantage of machine learning to identify aggressive drivers and prevent crash events. Aggressive drivers behave recklessly, including speeding, improper following, and risky lane-changing, and put people and property at risk. In recent years, many studies have worked on the method of driving pattern identification based on naturalist driving experiments [1,2,3], in which experiment vehicles were equipped with cameras to capture driver motions and the surrounding environment [4,5], and with specialized sensors to collect vehicle operation data and vehicle trajectory data [6,7]. Some research adopted driving simulators to detect driving behavior in the designed driving environment [8,9]. Compared to naturalistic driving study and driving simulator, video surveillance deployed on the roadside can provide a large amount of traffic environment data and vehicle trajectory data at a relatively affordable expense [10]. However, driving behavior extraction from the video can be time-consuming and computation-intensive [11,12,13]. Therefore, it is very important to find effective features based on the data extracted from the video to aid in driving pattern recognition.

Features that were used to recognize driving patterns can be grouped into three types: (1) control data, (2) physiological data, and (3) vehicle trajectory data. The in-vehicle cameras and sensors could capture the control data such as throttle opening and pedal brake, and wheel steering [2,3]. The images of the driver’s face and eye movement were also recorded to analyze distracted driving [3,4]. Some researchers adopted physiological data to identify drowsy driving. For example, Khushaba et al. [14] extracted drowsiness related data through electrocardiograph (ECG), electroencephalogram (EEG), and electrooculogram (EOG). Fu et al. [15] put electrodes on the drivers’ skin to collect physiological data. The control data and physiological data of drivers require additional cameras and sensors installment in the vehicle, which limits their application in practice, while vehicle trajectory data can be extracted from video surveillance. The vehicle trajectory data consists of lateral and longitudinal data. Murphey et al. [16] adopted longitudinal data, including speed, acceleration, and acceleration jerk, to classify drivers as aggressive, moderate, and calm. The lateral data, i.e., yaw rate, the distance from lane mark, the lateral acceleration, were also used to identify driving styles and driving patterns [9,17].

It is difficult to observe real traffic accident and target vehicle’s driving behavior before the accident. For example, Strategic Highway Research Program 2 (SHRP 2) Naturalistic Driving Study contains 41,478 records in total while there are only 102 crash records. Collision surrogate measurements are widely used to measure a vehicle’s risk level, such as Time to Collision (TTC), Headway, Deceleration Rate to Avoid Crash (DRAC), etc. [18,19]. Mahmud et al. [20] discussed the advantages and disadvantages of many existing proximal surrogate indicators of rear-end collision. TTC cannot handle zero relative speed properly in car-following and assume constant speed for both leading and following vehicles. Margin to collision (MTC), first proposed by Kitajima et al. [21], measures the crash potential in the case that both leading and following vehicles decelerate abruptly. Difference of Space distance and Stopping distance (DSS) [22], and similarly Potential Index for Collision with Urgent Deceleration (PICUD) [23] further considers the reaction time of deceleration for the following vehicle, and Time Integrated DSS (TIDSS) [24] calculates the aggregated crash risk for a vehicle by integrating the gap between DSS and the dangerous threshold value over a certain period.

Behavior recognition methods, especially machine learning algorithms, have been studied in many previous works. Different types of neural network (NN) algorithms such as back-propagation neural network (BPNN), multilayer feed-forward neural network (MLP), and constructive probabilistic neural network (CPNN) were adopted to identify driving behaviors [6,25,26]. Some studies proposed Hidden Markov Model (HMM) to detect dangerous driving behaviors [27], which could be challenging with a large number of states to be estimated [28]. SVM also has been widely applied to various kinds of pattern recognition problems, including voice identification, text categorization, and face detection [29,30].

Most machine learning algorithms were designed for balanced data, and class imbalance can impact the algorithm’s performance. Unfortunately, driving behavior data is usually imbalanced. The number of positive samples, for example, aggressive drivers are outnumbered by normal drivers in real-world traffic. Aggressive drivers as the minority in the dataset may be ignored by classification algorithms, which tend to focus on predicting normal drivers. Therefore, using imbalanced data in the training of machine learning algorithms may lead to biased results and bad performance on aggressive driver recognition. Two main solutions to imbalance classification problems are cost-sensitive learning and resampling. Cost-sensitive learning assumes that the misclassification costs of normal and aggressive drivers are different. Higher misclassification costs for the minority class can force the model to better predict aggressive drivers. Resampling methods oversample the minority samples or undersample the majority samples to make the training data more balanced. Existing research tried cost-sensitive learning, or different resampling methods as data preprocessing with no modification on the existing machine learning algorithms.

Different from existing studies, this paper implements imbalanced class boosting algorithms, such as SMOTEBoost [31], RUSBoost [32], and CUSBoost [33], which have not been applied to recognition driving behavior or driving style. Imbalanced class boosting algorithms apply the resampling method to increase the number of minority class samples or decrease the number of majority class samples in each iteration of boosting. Imbalanced class boosting algorithms were tested on many datasets, and they give better results than using the resampling method only as data preprocessing [31,32,33]. The recognition results of imbalanced class boosting algorithms are compared with cost-sensitive boosting and boosting with resampling.

In the paper, the Next Generation Simulation (NGSIM) vehicle trajectory data extracted from video surveillance is adopted to measure the rear-end collision risk for each driver. The advantage of video-extracted vehicle trajectory data is the huge number of vehicles that can be observed simultaneously with relatively low cost of video recording and video processing. A large sample of vehicle trajectory can help better train data-hungry machine learning models. Once the recognition model is well-trained, the identification of drivers can be done using any data source that provides vehicle trajectory information, including video-extracted data, in-vehicle sensor data, and cell-phone data. If the data used for training and identification are from different sources, then the multiple-source data should be calibrated to have consistent measurement accuracy.

Among many aggressive driving behaviors, improper following is the most common one in the NGSIM vehicle trajectory data and its severity can be quantified using the surrogate measurement of rear-end collision. In this paper, the driving aggressiveness is determined and labeled by a proposed surrogate measurement of rear-end collision, which can be calculated automatically using vehicle trajectory data and therefore is more efficient and objective than questionnaires and the subjective judgment of experts. Then different algorithms are applied to identify aggressive drivers with labeled and imbalanced data. The identification results are discussed based on performance indicators.

## 2. Materials and Methods

The methodology can be divided into two parts. The first part includes Section 2.1. and Section 2.2., which explains how we measure the collision risk of each vehicle and label aggressive drivers. In Section 2.1., we propose a new measurement of rear-end collision risk. In Section 2.2., we introduce three anomaly detection methods to determine the threshold of aggressive driving. The second part explains how to train a classification model once each driver has been labeled. Section 2.3. covers the feature extraction with the Discrete Fourier Transform (DFT) method, which transforms a given time series to signal amplitude in the frequency domain, which can reveal driving characteristics hidden in vehicle trajectory data. Section 2.4. introduces imbalanced class boosting algorithms and other algorithms tested in this paper. The last section shows the performance indicators used to measure the ability of boosting algorithms. The methodology framework is shown in Figure 1. This research does not involve human participants and the studied data contains no sensitive and personally identifiable information.

### 2.1. *Surrogate Measurement of Collision*

As discussed in the Introduction section, there are many proximal surrogate indicators proposed to measure the collision risk or evaluate safety level. Among all, Time to Collision (TTC) is commonly used in the collision warning system. It assumes constant driving speed for both leading and following vehicles and ignores the scenario that the leading vehicle decelerates abruptly, which may underestimate the crash risk of the following vehicle in unsteady traffic flow. Margin to Collision (MTC) overcomes the disadvantage of TTC by assuming that both leading and following vehicles can decelerate abruptly at the same time. Difference of Space distance and stopping distance (DSS) considers the following vehicle’s reaction time to the leading vehicle’s deceleration. Time Integrated DSS (TIDSS) calculates the integrate DSS over a time period to show the aggregated crash risk of a given vehicle.

We propose a new surrogate indicator, Average Crash Risk (ACR), to measure a driver’s aggressiveness. For each vehicle, Crash Risk (CR) at time point t is calculated based on its DSS:(1)$$DSS\left(t\right)=\frac{v_{l}^{2}\left(t\right)-v_{f}^{2}\left(t\right)}{2\mu g}+d\left(t\right)-\tau v_{f}\left(t\right)$$
where *v_l_* and *v_f_* are the speed of the leading and following vehicles, respectively, *μ* is the fraction rate, set to 0.7; *g* is the acceleration of gravity, 9.8 m/s^2^ (or 32.174 ft/s^2^); *d* is the gap between the leading and following vehicles; *τ* is the reaction time of drivers. When the vehicle is accelerating, *τ* is set to 1.5 s. When the vehicle is decelerating or idling, *τ* is set to 0.7 s.

When DSS > 0, it means the following vehicle has enough time to decelerate and avoid a collision. Therefore, Crash Risk is 0. When DSS ≤ 0, the following vehicle has a potential crash risk, and the Crash Risk is measured as the absolute value of DSS divided by the speed of the following vehicle.
(2)$$CR\left(t\right)=\{\begin{array}{cc} 0 & \mathrm{if}\;\mathrm{DSS}\left(t\right)\text{>}0\;\\ \left|DSS\left(t\right)\right|/v_{f}\left(t\right) & \mathrm{if}\;\mathrm{DSS}\left(t\right)\leq 0 \end{array}$$

To measure the overall driving aggressiveness for each driver during the whole car-following process, we calculate the average crash risk as follows:(3)$$\mathrm{ACR}=\frac{1}{T}\overset{T}{\underset{t=0}{\sum }}CR\left(t\right)\times \Delta t$$
where T is the car-following duration; Δ*t* is the sampling interval, 0.1 s.

### 2.2. Average Crash Risk (ACR) Threshold

Once each driver’s ACR is calculated based on how they interact with their preceding vehicle, aggressive drivers can be determined as their ACR exceeds a certain threshold. However, there is no empirical or theoretical threshold available in previous studies, and we apply three anomaly detection methods to find the boundary between normal drivers and aggressive drivers: K-means clustering, interquartile range rule, and Xth percentile.

#### 2.2.1. K-means clustering

Given a set of observations (x_1_, x_2_, …, x_n_), where each observation is a d-dimensional real vector, K-means clustering aims to partition the *n* observations into *k* groups = {C_1_, C_2_,…, C*_k_*} to minimize the within-cluster sum of variance. The objective of K-means is:(4)$$E=\overset{k}{\underset{i=1}{\sum }}\underset{x\in C_{i}}{\sum }x-\mu_{i}{}_{2}^{2}$$
where $\mu_{i}=\frac{1}{\left|C_{i}\right|}\sum_{x\in C_{i}}x$ is the mean vector within cluster Ci.

The K-means algorithm uses the squared Euclidean distance metric and a heuristic to find centroid seeds for *k*-means clustering. We use *k*-means to group 299 drivers into 2 clusters: normal and aggressive.

#### 2.2.2. Interquartile Range Rule

The interquartile range rule is useful in detecting outliers that fall far away from the center of the data. We assume that an aggressive driver’s ACR is extremely apart from the average and use the following equation to calculate the threshold.
(5)$${\mathrm{Threshold}=Q}_{3}+1.5\times \mathrm{IQR}$$
where Q_3_ is the 75th percentile of the data; IQR is the difference between the 75th percentile and the 25th percentile of the data.

#### 2.2.3. The Xth percentile

The Xth percentile method is straightforward. Using the Xth percentile of the data as the threshold is equivalent to assuming that (100−X)% of drivers on the road are aggressive. The proper value of X is vague and subjective. We take the 94th percentile of the ACR as the threshold and use it only as complementary to the other methods above.

### 2.3. Discrete Fourier Transform

Since every vehicle has a different length of trajectory data, the time series of gap, speed, and acceleration rate of each vehicle cannot be used directly to identify the driver’s driving aggressiveness. Discrete Fourier Transform (DFT) has been applied many times in driving behavior studies to convert time series of driving features to signal amplitude in the frequency domain.

The DFT of a given time series (x_1_, x_2_,…, x_N_) is defined as a sequence of N complex numbers $\left(DFT_{0},\;DFT_{1},\ldots ,\;DFT_{N-1}\right)$:(6)$$DFT_{k}=\overset{N-1}{\underset{n=0}{\sum }}x_{n}e^{\left(-\frac{2\pi i}{N}kn\right)}$$
where *i* is the imaginary unit.

Time series data have temporal structures. The low/medium-frequency information shows the time series’ level, trend, and periodicity. The high-frequency part is noise. Through Discrete Fourier Transform, we keep the first 15 DFT coefficients of each time series as input and drop the rest part as noise.

### 2.4. Imbalanced Class Boosting Algorithms

In real traffic, the proportion of aggressive drivers is much smaller than the proportion of normal drivers. Therefore, the dataset is usually imbalanced with aggressive drivers as the minority class. A popular solution is to fully or partially balance the class distribution by resampling. For example, SMOTE (Synthetic Minority Oversampling Technique) [34] balances the data by synthetically generating more instances of the minority class, and the classifiers can broaden their decision regions for the minority class. RUS (Random Under Sampling) removes examples from the majority class at random until the desired class distribution is achieved.

The first imbalanced class boosting algorithm SMOTEBoost was proposed by Chawla et al. [31]. It combines the SMOTE and the standard boosting procedure. The standard boosting procedure gives equal weights to all misclassified examples, and sampling distributions in subsequent boosting iterations could have a larger composition of majority class cases. By introducing SMOTE in each round of boosting, SMOTEBoost algorithm gradually increases the number of minority class samples. The algorithm procedure of AdaBoost and imbalanced class boosting are shown in Figure 2. Seiffert et al. [32] proposed the RUSBoost algorithm that combines RUS and AdaBoost. In each iteration of boosting, RUS is used to balance class instead of SMOTE. CUSBoost [33] is another imbalanced class boosting algorithm that combines under-sampling with AdaBoost. CUSBoost clusters majority class first, and then randomly removes majority samples based on their cluster.

Nine algorithms are tested in the paper (see Table 1). The first group is cost-sensitive boosting, including AdaBoost and XGBoost, which does not resample the training data. Instead, a higher-class weight was set for the minority class to offset the imbalance. The second group is standard boosting with resampling. We tried two resampling methods: SMOTE and RUS. There are four algorithms in the second group. SMOTE + AdaBoost, for example, uses SMOTE first on the training data to oversample the minority class, and then train the AdaBoost model using the balanced training data. The third group is imbalanced class boosting, including SMOTEBoost, RUSBoost, and CUSBoost. Hyperparameter optimization for each machine learning model is achieved by Grid Search.

### 2.5. Performance Evaluation

The performance of the boosting algorithm depends on its power to identify aggressive drivers using vehicle trajectory data. This paper use four important performance indices: precision rate, recall rate, f1 score, and Area under the Precision-Recall Curve (AUPRC).

Precision rate is defined as follows:(7)$$\mathrm{Precision}=\frac{TP}{\left(TP+FP\right)}$$
where *TP* is the number of aggressive drivers correctly identified; *FP* is the number of normal drivers wrongly identified as aggressive drivers.

Recall rate is defined as follows:(8)$$\mathrm{Recall}=\frac{TP}{\left(TP+FN\right)}$$
where *FN* is the number of aggressive drivers wrongly identified as normal drivers.

The F1 score is the harmonic average of the precision and recall. A high F1 score represents high values in both precision rate and recall rate.
(9)$$F1=2\frac{\mathrm{Precision}\;\times \;\mathrm{Recall}}{\mathrm{Precision}+\mathrm{Recall}}$$

A ROC curve (receiver operating characteristic curve) is a graph showing the false positive rate versus the true positive rate for different candidate threshold values between 0 and 1. Similarly, a precision-recall curve is a plot of the precision and the recall for different thresholds. Generally, ROC curves should be used when there are roughly equal numbers of observations for each class. When there is a class imbalance, Precision-Recall curves should be used, because the ROC curve with an imbalanced dataset might be deceptive and lead to incorrect interpretations of the model skill [35]. Therefore, this paper uses AUPRC to compare algorithms’ performance, which measures the entire two-dimensional area underneath the entire Precision-Recall curve.

Stratified K-fold cross-validation is widely used to evaluate the classification algorithm’s performance, especially when the dataset is highly imbalanced. Since using all 299 drivers to train the model may cause an overfitting problem and exaggerate the accuracy of the trained model, we divided the 299 drivers randomly into five equal-sized subsets. At each time, four subsets are used for resampling and then training, and the left-out subset is used to assess the performance of the trained model. This process rotates through each subset, and the average accuracy, precision rate, and recall rate represent the performance of the algorithm. Stratified 5-fold cross-validation was repeated five times.

## 3. Data

Next Generation Simulation (NGSIM) is the most researched public dataset of vehicle trajectory. It was originally collected using cameras, and vehicle trajectory data was extracted through computer vision techniques. One part of NGSIM data was collected on a segment of the I-80 freeway in Emeryville, California. The segment is approximately 500 m in length and contains six lanes, including a high occupancy vehicle (HOV) lane. The data were collected from 4:00–4:15 pm and from 5:00–5:30 pm on 13 April 2005, 45 min in total. Due to low-resolution cameras and mistracking of vehicles from video images, the NGSIM trajectory data has substantial measurement error [36,37]. Montanino and Punzo [36] reconstructed the I-80 dataset 1 (from 4:00–4:15 pm), which was shown significant improvement over the original NGSIM dataset. The reconstructed NGSIM data is available to the public on the U.S. Department of Transportation’s public data portal website.

This paper uses Montanino and Punzo’s dataset [36] for aggressive driver identification and focuses on 299 leader-follower vehicle pairs (LVP) on the HOV lane that was not interrupted by lane-changing. Each leader-follower pair has a duration of at least 10 s, most are longer than 20 s. For every 0.1 s, the leading and following vehicles’ speed, acceleration/deceleration rate, and gap were recorded. With the trajectory data of 299 LVPs, we can identify the 299 following driver’s driving aggressiveness based on their interaction with the leading vehicle.

Since the data was collected during peak hours, the aggressive driving recognition model developed in this paper may not be suitable for nonpeak hours and weekends. Shinar and Compton [38] found that the likelihood of aggressive driving during peak hours is higher than nonpeak hours and weekends because drivers have higher values of time in peak hours and then more motives to drive recklessly. Congestion may also trigger anger, frustration, and depression and lead to angry and aggressive driving.

## 4. Result I: Driving Aggressiveness Labeling

### 4.1. Average Crash Risk Threshold

Using the NGSIM trajectory data of 299 LVPs, we can calculate the Average Crash Risk (ACR) for each following vehicle in LVPs. ACR reflects the driver’s aggressiveness in the car-following process. Figure 3 shows the histogram of Average Crash Risk for all 299 drivers. In total, 35.7% of drivers have an ACR = 0, which means these drivers never have negative DSS value and always have enough time to avoid a collision, and 65.8% of drivers have an ACR < 0.1, which indicates only occasional and temporary crash risk during the car-following process.

Table 2 shows three different ACR thresholds, each determined by a method discussed in Section 2.2. K-means clustering algorithm generates the smallest ACR threshold (denoted as ACR_1_), 0.14. If labeling drivers with ACR higher than ACR_1_ as aggressive drivers, then out of 299 drivers, there are 43 aggressive ones and 256 normal ones. Aggressive drivers account for 14.4% of the driver population. The ACR threshold is 0.19 under the “interquartile range rule” (denoted as ACR_2_). The 94th percentile of ACR distribution is 0.28 (denoted as ACR_3_). The k-means clustering method labels 14.4% drivers as aggressive, and then create a less imbalanced dataset. Because ACR_1_ < ACR_2_ < ACR_3_, the “interquartile range rule” method and “Xth percentile” method label fewer drivers as aggressive, and therefore they increase the imbalance ratio to 9:1 and 14:1, respectively.

With different combinations of inputs and threshold values, we generate five datasets of driver aggressiveness. Dataset 1-3 have the same ACR threshold value and different input features. Dataset 3–5 have the same input features and different imbalance ratios. Comparison of results from datasets 1–3 can show us the importance of input features, and comparison of results from datasets 3–5 can demonstrate the impact of imbalance ratio on the performance of classification boosting algorithms.

### 4.2. Crash Risk and Driving Aggressiveness

There is one question we need to answer before using ACR to measure driving aggressiveness: is a high crash risk equivalent to aggressive driving? Several external factors may increase a vehicle’s crash risk. For example, unexpected and abrupt acceleration and deceleration by the leading vehicle, either caused by traffic congestion or the leading vehicle’s reckless driving style, may lead to high crash risk for the following vehicle. If the following vehicle is not responsible, or not completely responsible for the high crash risk, then using ACR to measure driving aggressiveness could be problematic.

To rule out the possibility that traffic condition and the leading vehicle’s driving style may impact the following vehicle’s crash risk, we calculated the correlation between the leader and follower’s ACRs based on a simple logic: if traffic condition has an impact on vehicle’s ACR, then both the leading and the following vehicle in the same traffic flow should have a similar crash risk level. If the leading vehicle’s reckless driving has an influence on the following vehicle, making it more aggressive or more conservative, then the ACRs of two vehicles should also show some degree of correlation.

Among 299 leader-follower pairs (LVP), 264 leading vehicle’s driving aggressiveness are also determined since they are following vehicles in other LVP. Therefore, we can calculate the correlation of ACR for these 264 LVPs. The Pearson correlation coefficient between the leader and follower’s ACRs is calculated as 0.01, which indicates that the leading and following vehicles’ ACR are nearly independent, and the leading vehicle’s crash risk is not delivered to its follower. Therefore, it implies that a driver’s ACR is impacted by the leading vehicle and can represent the driver’s aggressiveness.

## 5. Result II: Driving Aggressiveness Recognition

### 5.1. The Performance of Boosting Algorithms

After five times of 5-fold cross-validation, the average precision rate, recall rate, f1 score, and AUPRC of each algorithm are posted in Table 3, Table 4 and Table 5.

We trained nine algorithms with different sets of inputs. First, we only used the DFT coefficients of speed and acceleration from 0–1.5 Hz. Table 3 shows that CUSBoost generates the best AUPRC 0.715, outperforming XGBoost, whose APRC is 0.695. However, CUSBoost’s precision rate and recall rate are not so high. RUSBoost gives the highest recall rate for aggressive drivers, 92.8%, which means only 7.2% of aggressive drivers are misclassified as normal drivers. Most precision rates are low, except XGBoost, which gives the highest precision rate as 80.9%, which means in all drivers being identified as aggressive drivers, 19.1% of which are “false-alarm”. XGBoost has the highest F1 score of 0.639, followed by CUSBoost, due to its high performance in both precision rate and recall rate.

In the second dataset, we introduce the DFT coefficients of the gap between Leading-following Vehicle Pairs into the input. The results are shown in Table 4. Using the gap as the input, instead of speed and acceleration, can significantly improve the ability of recognition. The highest AUPRC is 0.917, achieved by XGBoost. CUSBoost has the second highest AUPRC 0.912. RUSBoost gives the highest recall rate for aggressive drivers, 96.2%, which means only 3.8% of aggressive drivers are misclassified as normal drivers. XGBoost gives the highest precision rate as 91.0%, which means in all drivers being identified as aggressive drivers, 9.0% of which are “false-alarm”. SMOTE + XGBoost has the highest F1 score of 0.903, followed by XGBoost and CUSBoost.

In the third dataset, we combine the DFT coefficients of vehicle speed, acceleration, and gap all together as the input. Table 5 shows that SMOTEBoost gives the highest AUPRC of 0.942, while RUSBoost gives the highest recall rate, 0.954. RUS + XGBoost gives the highest Precision rate and F1 score.

### 5.2. The Impact of Imbalance Ratio

Figure 4 shows the AUPRC values each algorithm achieved in Dataset 3–5. We find that SMOTEBoost and CUSBoost are robust with a highly imbalanced dataset. SMOTEBoost and CUSBoost have AUPRC higher than 0.9 in all three datasets. When imbalance ratio increases from 6:1 (Dataset 3) to 14:1 (Dataset 5), AdaBoost and XGBoost’s AUPRC declines significantly, even with resampling. SMOTE + AdaBoost and RUSBoost can get better AUPRC when imbalance ratio increases; however, their performance is not stable. SMOTE + AdaBoost has the lowest AUPRC among all algorithms in Dataset 3. RUSBoost has the second-lowest AUPRC in Dataset 4.

### 5.3. The Impact of Resampling

We compare the difference of AUPRC between cost-sensitive boosting without resampling and standard boosting with resampling in Figure 5 and Figure 6. By balancing the majority class and the minority class in the train data with resampling, we expect that AdaBoost and XGBoost can be fitted to predict aggressive drivers better and then have a higher AUPRC. However, the result is not consistent. On one hand, using Random Under Sampling almost always pushes down AUPRC, compared to cost-sensitive learning. On the other hand, using SMOTE resampling slightly raises AdaBoost’s AUPRC in Dataset 2 and Dataset 5, but reduces AdaBoost’s AUPRC in Dataset 1, 3, and 4. In all datasets, it is better to train cost-sensitive XGBoost model directly without using any resampling techniques. Shown in Figure 5, SMOTE + XGBoost and RUS + XGBoost’s AUPRC are always lower than XGBoost’s. 

## 6. Discussion

### 6.1. ACR and Aggressiveness

There is one question we need to answer: is a high crash risk equivalent to aggressive driving? There are several external factors that may impact a vehicle’s crash risk. For example, frequent/abrupt acceleration and braking by the leading vehicle, either caused by unstable traffic flow or the leading vehicle’s reckless driving style, may lead to high crash risk for the following vehicle. If the following vehicle is not responsible, or not completely responsible for the high crash risk, then using ACR or any rear-end collision surrogate indicator to measure driving aggressiveness could be problematic.

To rule out the possibility that traffic condition and/or the leading vehicle’s driving style may impact the following vehicle’s crash risk, we calculated the correlation between the leader and follower’s ACRs based on simple logic. If traffic condition has an impact on vehicle’s ACR, then both leading and following vehicle in the same traffic flow should have similar crash risk level. If the leading vehicle’s reckless driving has an influence on the following vehicle, making whom more aggressive or more defensive, then the ACRs of two vehicles should also show some degree of correlation.

Among 299 leader-follower pairs (LVP), 264 leading vehicle’s driving aggressiveness are also determined since they are following vehicles in other LVP. Therefore, we are able to calculate the correlation of ACR for these 264 LVPs. The Pearson correlation coefficient between the leader and follower’s ACRs is −0.0137, which indicates that the leading vehicle’s crash risk is not delivered to its follower, and also implies that ACR can represent driving aggressiveness.

### 6.2. Algorithm Performance

Based on the results shown in Table 3 to Table 5, we find that imbalanced class boosting algorithms, SMOTEBoost and CUSBoost, generally outperform other boosting algorithms. XGBoost also performs well when the imbalance ratio of the dataset is moderate.

The advantage of the imbalanced class boosting algorithm is more obvious with a high imbalance ratio. For example, in Dataset 3, aggressive drivers account for 14.4% of all drivers, and the performance difference between XGBoost, which is the best cost-sensitive boosting algorithm, and CUSBoost is small (AUPRC: 0.938 vs. 0.935). By contrast, in Dataset 5, in which only 6.4% of drivers are aggressive, the AUPRC of XGBoost decreases from 0.938–0.871, while the AUPRC of CUSBoost only drops slightly from 0.935–0.924.

It is surprising to find that SMOTE + AdaBoost and RUS + AdaBoost give worse results than AdaBoost, and SMOTE + XGBoost and RUS + XGBoost give worse result than XGBoost. Several existing studies used SMOTE or other resampling methods before boosting, assuming that it will create better results for imbalanced data. It implies that apply the resampling method before boosting algorithms does not guarantee a better recognition result than cost-sensitive learning. One possible reason is that a one-time resampling in the training data before the boosting may skew the data distribution and later weaken the boosting algorithm’s power to recognize the test data because test data is not balanced by the resampling method.

### 6.3. Mode Input

We found that using the discrete Fourier coefficients of acceleration alone as the input was much worse than using other inputs. Due to acceleration’s poor ability to recognize aggressive drivers, its result is not included in this paper. It is a little surprising since some previous studies found that acceleration, pedal brake, or throttle opening have significant power in driving style classification. For example, Kluger et al. [1] found a distinct difference in acceleration discrete Fourier coefficients between vehicles with safety-critical events and vehicles at baseline. There are several possible explanations. First, acceleration is the first-order derivative of speed, which implies that the information of acceleration has been included in the DFT coefficient of speed. Second, the trajectory data in the paper was recorded on a highway near night peak-hour, the acceleration pattern would be different from vehicles in free-flow traffic, local street traffic, or vehicle data generated from driving simulator with no traffic or light traffic. Third, the fidelity of acceleration in NGSIM, even after reconstruction by Montanino and Punzo [34], is still the second-order derivative of vehicle position extracted from the video. Errors in vehicle position will be amplified and delivered to vehicle acceleration. The innate disadvantage of video-based trajectory data might be the reason for the indifference between normal and aggressive driver’s acceleration DFT coefficients.

## 7. Conclusions

The objectives of this research are mainly three things: find out how to label drivers using a new collision surrogate measurement, how to identify aggressive drivers using imbalanced class boosting, and what is the key feature.

This paper takes advantage of the reconstructed NGSIM trajectory data to explore the possibility of identifying aggressive drivers. To label each driver’s driving aggressiveness, we propose a surrogate measurement of collision risk, Average Crash Risk (ACR), which distinguishes aggressive drivers from others based on their response in the car-following process. Compared to other labeling methods, like experts’ subjective judgment, questionnaire, or clustering based on speed/acceleration/wheel steering, surrogate measurement is more suitable in real-world traffic. The correlation of collision risk between leading and following vehicles is tested. This paper found that the crash risk of the leading vehicle, measured by ACR proposed in this paper, has no impact on the following vehicle, and the driving aggressiveness of the two drivers are independent.

We found the gap between the leading and following vehicles is the key feature to recognize aggressive drivers. A vehicle’s speed and acceleration can be influenced by its leading vehicle, and we found using speed and acceleration alone cannot identify the driver’s aggressiveness with acceptable precision rate and recall rate. By contrast, using the gap alone as the input can train a model with precision rate and recall rate both at a 90% level.

Imbalanced class boosting algorithms show their ability to handle imbalanced driving data. The more imbalanced the data is, the more necessary it is to use an imbalanced class boosting algorithm rather than a standard classification algorithm. When using the discrete Fourier coefficients of the gap, speed, and acceleration as the input features, SMOTEBoost, RUSBoost, and CUSBoost outperform AdaBoost and XGBoost in the most imbalanced Dataset 5. Resampling imbalanced data before AdaBoost or XGBoost does not always improve the model’s recognition ability. Since SMOTEBoost, CUSBoost, and RUSBoost are modified AdaBoost with the resampling method, their performance can be further improved by replacing AdaBoost with more advanced boosting algorithms.

## Figures and Tables

**Figure 1 ijerph-17-02375-f001:**
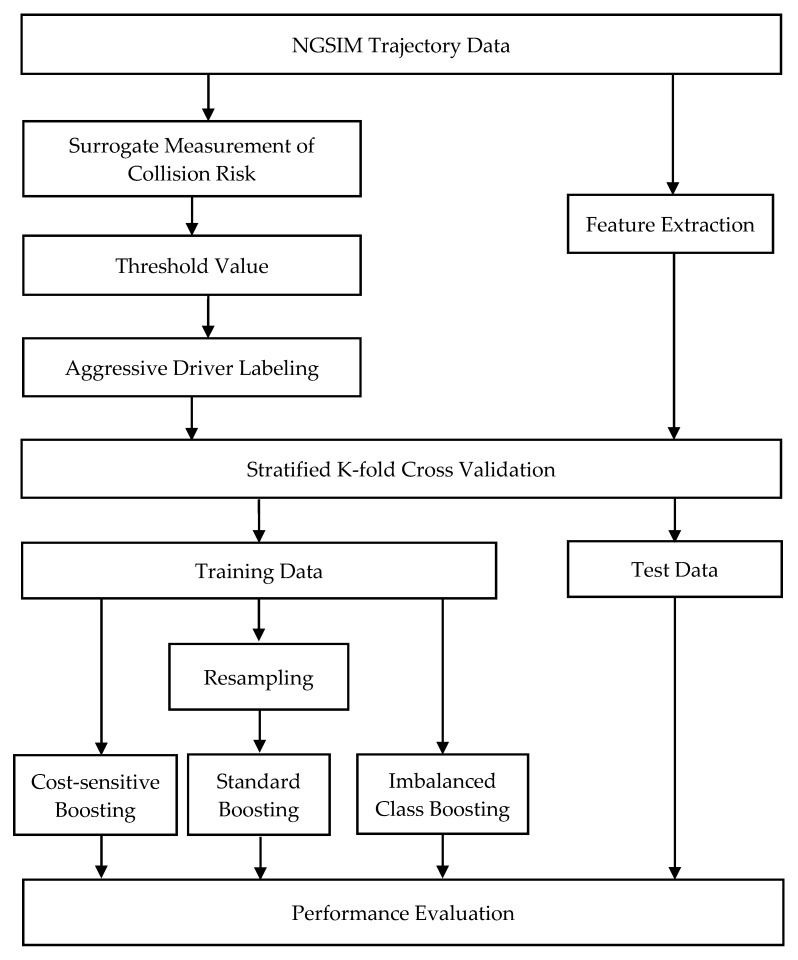
Methodology framework.

**Figure 2 ijerph-17-02375-f002:**
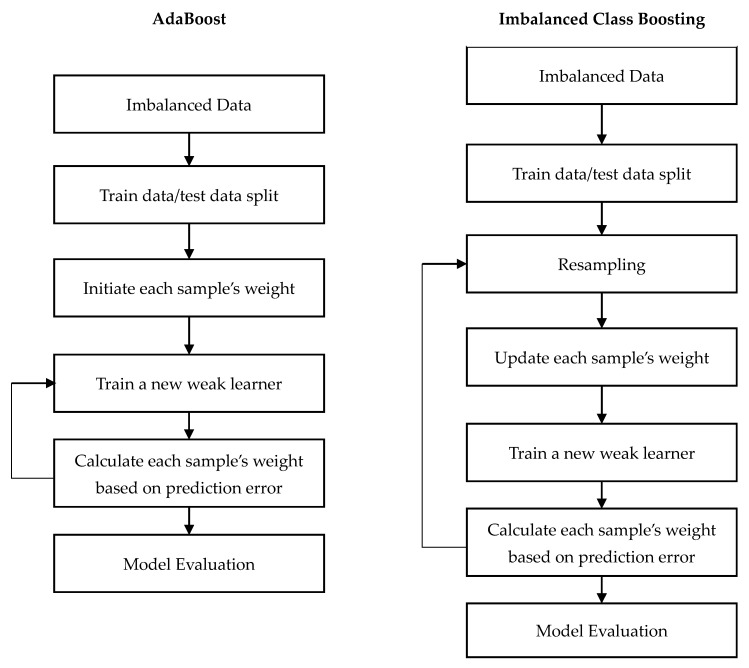
Difference in procedure between AdaBoost and imbalanced class boosting.

**Figure 3 ijerph-17-02375-f003:**
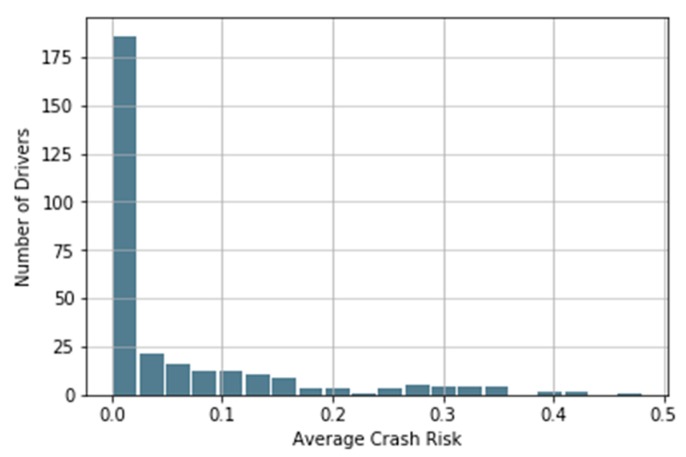
Histogram of Average Crash Risk (ACR).

**Figure 4 ijerph-17-02375-f004:**
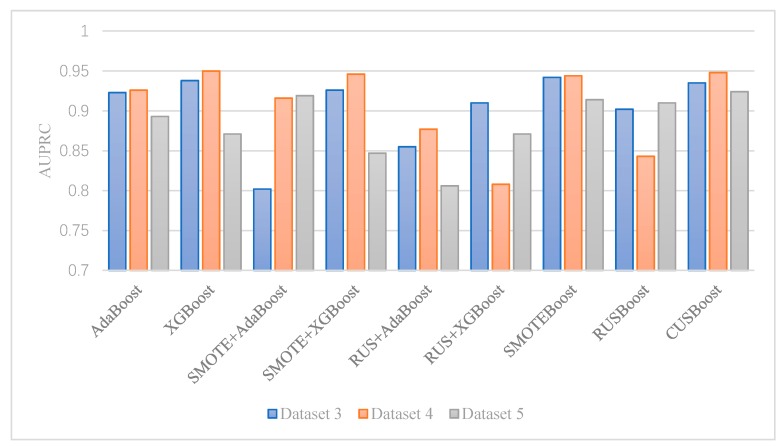
The impact of imbalance ratio on AdaBoost.

**Figure 5 ijerph-17-02375-f005:**
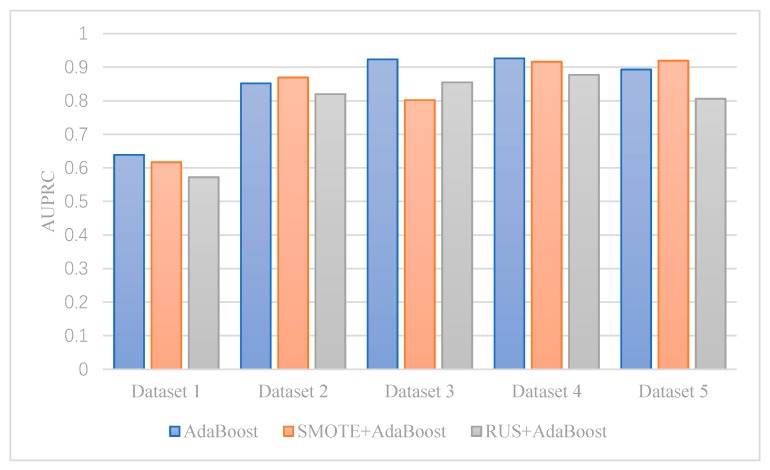
The impact of resampling on AdaBoost.

**Figure 6 ijerph-17-02375-f006:**
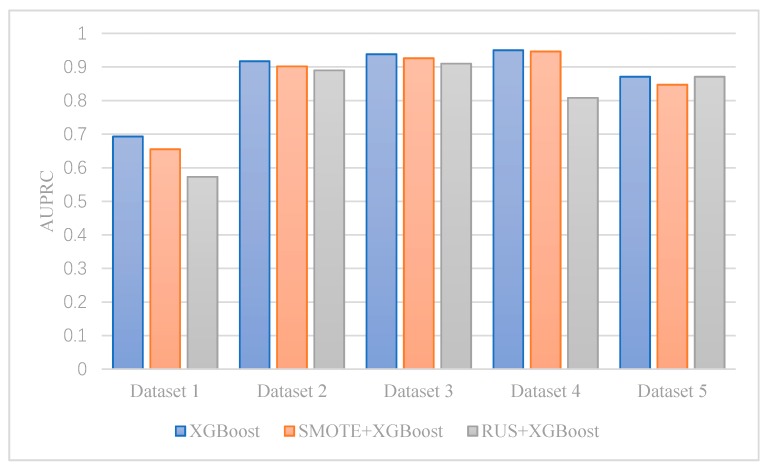
The impact of resampling on XGBoost.

**Table 1 ijerph-17-02375-t001:** Nine algorithms tested.

**Boosting Algorithm**	**Resample Training Data**
Cost-sensitive boosting	
AdaBoost	No
XGBoost	No
Standard boosting with resampling	
SMOTE + AdaBoost	SMOTE
SMOTE + XGBoost	SMOTE
RUS + AdaBoost	Random undersampling
RUS + XGBoost	Random undersampling
Imbalanced Class Boosting	
SMOTEBoost	No
RUSBoost	No
CUSBoost	No

**Table 2 ijerph-17-02375-t002:** Characteristics of datasets.

**Dataset**	**Input**	**Method**	**ACR Threshold Value**	**Percentage of Aggressive Drivers**	**Imbalance Ratio**
Dataset 1	DFT of speed and acceleration	K-means clustering	0.14	14.4%	6:1
Dataset 2	DFT of gap	K-means clustering	0.14	14.4%	6:1
Dataset 3	DFT of speed, acceleration, and gap	K-means clustering	0.14	14.4%	6:1
Dataset 4	DFT of speed, acceleration, and gap	Interquartile range rule	0.19	10.0%	9:1
Dataset 5	DFT of speed, acceleration, and gap	94th percentile	0.28	6.4%	14:1

**Table 3 ijerph-17-02375-t003:** The performance of boosting algorithms (Dataset 1).

Algorithms	Precision	Recall	F1 Score	AUPRC
Cost-sensitive boosting				
AdaBoost	0.720	0.504	0.561	0.639
XGBoost	0.809	0.552	0.639	0.693
Standard boosting with resampling				
SMOTE + AdaBoost	0.495	0.663	0.557	0.617
SMOTE + XGBoost	0.526	0.684	0.586	0.655
RUS + AdaBoost	0.414	0.763	0.529	0.572
RUS + XGBoost	0.432	0.779	0.551	0.573
Imbalanced class boosting				
SMOTEBoost	0.441	0.823	0.571	0.664
RUSBoost	0.297	0.928	0.445	0.507
CUSBoost	0.586	0.661	0.615	0.715

**Table 4 ijerph-17-02375-t004:** The performance of boosting algorithms (Dataset 2).

Algorithms	Precision	Recall	F1 Score	AUPRC
Cost-sensitive boosting				
AdaBoost	0.832	0.768	0.786	0.852
XGBoost	0.910	0.894	0.897	0.917
Standard boosting with resampling				
SMOTE + AdaBoost	0.845	0.824	0.825	0.869
SMOTE + XGBoost	0.887	0.930	0.903	0.902
RUS + AdaBoost	0.681	0.901	0.774	0.820
RUS + XGBoost	0.823	0.917	0.861	0.890
Imbalanced class boosting				
SMOTEBoost	0.799	0.856	0.818	0.895
RUSBoost	0.588	0.962	0.722	0.851
CUSBoost	0.840	0.908	0.866	0.912

**Table 5 ijerph-17-02375-t005:** The performance of boosting algorithms (Dataset 3).

Algorithms	Precision	Recall	F1 Score	AUPRC
Cost-sensitive boosting				
AdaBoost	0.890	0.824	0.847	0.923
XGBoost	0.924	0.893	0.904	0.938
Boosting with resampling				
SMOTE + AdaBoost	0.830	0.873	0.732	0.802
SMOTE + XGBoost	0.848	0.916	0.912	0.926
RUS + AdaBoost	0.827	0.888	0.806	0.855
RUS + XGBoost	0.925	0.929	0.914	0.910
Imbalanced class boosting				
SMOTEBoost	0.812	0.917	0.852	0.942
RUSBoost	0.605	0.954	0.730	0.902
CUSBoost	0.870	0.911	0.884	0.935

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
