# Peer review of "Improve Aggressive Driver Recognition Using Collision Surrogate Measurement and Imbalanced Class Boosting"

_ijerph, 2020, doi:10.3390/ijerph17072375_

Round 1

Reviewer 1 Report

Thanks to the author for the high-quality response. I still think labeling and predicting driver categories in a loop is very weird. But I agree with the author's point that fewer features will help reduce the use of sensors. Maybe it sounds like the lack of sensors is not a problem in modern times. Such thinking may be useful in certain restricted scenarios. Collecting data is always difficult for researchers.

In addition, this paper makes a useful exploration for the automated identification of aggressive driving. The imbalanced class boosting algorithm may play a more important role in future research. I think your work is worth reading. Hope to see more progress in the future.

Reviewer 2 Report

Dear Authors,
I sincerely appreciate the effort you have made to improve the article as directed by the reviewers. I believe that the article presents an adequate discussion of a problem that is occurring in our society.
In my opinion the article is publishable, correctly structured and leads the reader/researcher clearly through the document.
I encourage the authors to continue in this line of research, achieving more parameters that expand the approach and development of this topic.
Greetings

This manuscript is a resubmission of an earlier submission. The following is a list of the peer review reports and author responses from that submission.

Round 1

Reviewer 1 Report

- Page 1 Line 30:
The author introduces several aggressive driving behaviors here, but none of them seems to have been discussed in this study except for the improper following. What about other aggressive driving behaviors? When identifying whether a driver is aggressive, is it reasonable to only consider the improper following?

- Page 3 Line 89, "...In each round of boosting, the resampling method is used to increase the number of minority class samples.":
CUSBoost and RUSBoost are undersampling on majority class during resampling, instead of oversampling on minority samples.

- Page 2 Line 44:
The author divides the related studies into three categories, but does not explain why the vehicle trajectory was selected for this study. Is identification based on vehicle trajectory more effective than the other two kinds or can it solve different problems? In addition, is it possible to identify aggressive driving by fusing data from multiple sources?

- Page 3 Line 105, "...Discrete Fourier Transform... which can reveal driving characteristics hidden in vehicle trajectory data.":
As far as I know, time-frequency transformation is just changing the representation of data. Not sure what is the "characteristics hidden in vehicle trajectory data" here. There seems to be no further explanation for this later.

- Page 5 Line 133:
In the definition, CR is related to the spatial distance and the stopping distance. I agree with this in part. However, when there are two drivers and both of them maintain a space distance greater than the stopping distance, one is more safety-oriented than the other, thus keeping a larger distance from the front car. In this case, are their CR equal and both 0?

- Page 6 Line 172:
LVP definition in Page 9 Line 254 should be moved here.

- Page 9 Line 267:
What is the relationship between clustering and the ACR threshold? Is the threshold calculated by the quantile based on the aggressive cluster? Also, here are my **main** doubts about this research:

The author first generated two clusters of aggressive and normal drivers through k-means. Then a threshold was drawn based on the clustering results. Next, the drivers were classified as aggressive and normal again based on this threshold.

I am not sure if my understanding is accurate. If it is, the author has obtained the result of whether the driver is aggressive after clustering, why should the following tasks be carried out (including calculating thresholds, dividing aggressive and normal drivers again, and using several other models to make predictions)?

Similarly, if we already have a threshold to identify whether the driver is aggressive, why do we need to use another method to predict the result based on this threshold?

If a threshold or an algorithm is used to label data, then using the labeled data for prediction is essentially predicting the threshold or fitting the algorithm. In this paper, this threshold is ACR threshold, further speaking, it is a quartile or percentile derived from a specific algorithm. It is hard to say what is the meaning of such data processing.

This is a common problem in similar researches. When we label data with a certain rule and then use another model to predict these data, the best result of the model is to learn the labeling rule itself. This is also the reason why many related researches have to label data based on expert experience. Because we want to find an algorithm that can describe the experience of experts, instead of defining and predicting data categories in a loop.

- Page 10 Line 287 Table 2:
Why are there different criteria or threshold for judging aggressive drivers? What is the role of different thresholds in this part?

- Page 11 Line 322 Table 3:
In the table, we can find that the AdaBoost and XGBoost degrade after resampling. The same problem appears many times in the table below. Although the authors are aware of this problem, they do not seem to give an explanation.

- Page 15 Line 382 "...resampling methods before boosting, assuming that it will create better results for imbalanced data.":
In most unbalanced predictions cases, the main purpose of resampling is not to improve performance, but to avoid models with accuracy as the optimization objective ignoring categories with fewer samples. This is very important for anomaly detection tasks.

Accuracy is the proportion of samples (whether positive or negative) correctly classified, which is different from Precision, Recall or F1 score. When you use F1 score or AUC as the optimization target, the problem of minority class being ignored has been alleviated.

I suggest that the author should rewrite the method part. It is strongly recommended to have a comprehensive and accurate understanding of a method before using it.

Reviewer 2 Report

Why do you have a larger format for the Mathematical formulas?  some parameters (e.g., tau)  or characters with subscript and supercripts  do not line up . 

Reviewer 3 Report

Dear authors,

I think the approach in the article "Improve Aggressive Driver Recognition using Collision Surrogate Measurement and Imbalanced Class Boosting" is very good. I find a proposal with a suitable starting point and which addresses a new way of approaching studies of this type. The article as a whole is well developed, but I find some points that could be improved.

I think that the summary does not adequately reflect the objective of the research, as the article focuses much more on the optimization and use of the algorithms, rather than on the identification of aggressive drivers, or at least the discussion or conclusions are not sufficiently clear to me.

The introduction could include some comment on recent technologies in the car that can prevent reckless or dangerous behaviour, such as emergency braking when detecting obstacles or other vehicles. These tools prevent driving too close to vehicles in front of us. Although it is far from the objective and meaning of the article, I think it can be commented on as an element of reducing the risk of collision.

I think the paragraph between lines 79 and 85, which comments on the limitations of other studies where normal drivers are the norm, is very appropriate.

I believe that the paragraph between lines 86 and 92 should be developed a little further, since it leads to the contribution made in the article. I believe that the last part of the introduction could be improved a little in order to put the proposed method into a little more context before explaining it in point 2.

The methodological development is extensive and somewhat complex for those who do not understand it. A lot can be learned from this development, and it is well structured, clearly differentiating the three methods used.

They should be careful with the use of acronyms that have not been previously defined in the text, as happens, if I am not mistaken, in line 175 (LVP)

Within point "3" data, it would be interesting to include some reference to the influence that time of day can have on the classification of drivers according to their type of driving. Driving modes are not the same on a working day as on a public holiday, or according to the time of day. However, it is really well explained which time zone has been used.

On line 267 to 271, "Drivers with ACR higher 268 than 0.14 are labeled as aggressive drivers, while others as normal drivers", is a somewhat risky statement depending only on these parameters, it should be specified with less certainty.

This is not very important, but I think it should be fixed. In tables 3, 4 and 5, the figures are shown with 3 decimals, that is correct. All the amounts should have three decimal places, even if the last figure is a 0. In table 3, the number 0.720, instead of 0.72. Same case in table 4 and table 5.

In summary, the article is more inclined to a comparison of the application of the algorithms for the case of identification of aggressive drivers, more than in other aspects that would surround this phenomenon. The initial data appear to be correct, although in principle it seems that the analysis of driving modes will be more extensive.
